# The Role of TORS in the Management of Benign Pathology of the Base of Tongue: A Systematic Review

**DOI:** 10.3390/diagnostics15010005

**Published:** 2024-12-24

**Authors:** Riccardo Nocini, Valerio Arietti, Athena Arsie, Erica Zampieri, Luca Sacchetto

**Affiliations:** Unit of Otolaryngology, Head and Neck Department, University of Verona, P.le L.A. Scuro 10, 37134 Verona, Italy; riccardo.nocini@aovr.veneto.it (R.N.); athena.arsie@studenti.univr.it (A.A.); erica.zampieri@studenti.univr.it (E.Z.); luca.sacchetto@aovr.veneto.it (L.S.)

**Keywords:** TORS, base of tongue, oropharynx, surgery

## Abstract

Objective: Transoral robotic surgery (TORS) is becoming increasingly popular in head and neck surgery. Its applications have expanded beyond oncologic indications to obstructive sleep apnea syndrome (OSAS) and, more recently, to benign pathologies. Data Sources: A systematic search for articles published in the PubMed and Google Scholar databases between January 2003 and December 2023 was performed using the following combined search query (robot OR sleep OR apnea OR syndrome) AND (robot OR tongue OR base). Review methods: Given the limited literature, we conducted a systematic review focusing on the outcomes of TORS for benign pathologies of the base of the tongue. Our search methodology followed the Preferred Reporting Items for Systematic Reviews and Meta-Analyses (PRISMA) guidelines. Results: We found 16 articles that met our inclusion criteria. These were mainly case reports and a few case series. Conclusions: Compared to other transoral techniques, TORS offers better exposure, visualization, and access to the oropharynx, especially the base of the tongue, even in benign pathology. TORS should be considered a feasible, safe, and effective technique. Several more studies are needed to effectively evaluate the role of TORS in benign pathology that does not correlate with OSAS.

## 1. Introduction

Since the Da Vinci Surgical System was approved by the Food and Drug Administration (FDA) in 2009, it has become increasingly widespread. Over the years, new systems have been developed and approved [1]. Several techniques have been described for treating lesions at the base of the tongue, a hidden and sometimes very difficult-to-access area. The base of the tongue is a challenging region to access trans orally due to its anatomical location behind the main body of the tongue. This area tends to collapse quickly and contains recesses such as the glossoepiglottic fold, which are difficult to reach. The transoral route offers a limited view of this region, as the operator can observe only in a straight line. While open procedures have been considered for oncologic reasons, this is not the case for benign pathologies where the invasiveness of an open approach is often not justified. Open surgical approaches to the oropharynx can be associated with morbidities such as cosmetic deformity, malocclusion, and dysphagia. With the advent of TORS, robotic surgery is increasingly used in oncologic surgery of the base of the tongue to simplify treatment and be minimally invasive. The same is true for benign pathology, where better outcomes, or rather, do not justify the cost and time for preoperative preparation; there is no evidence for this in the literature. The TORS approach is indicated for different anatomical subgroups [2]. For the oropharynx, traditional transoral approaches using a cold blade, electrocautery, or laser-assisted endoscopic/microscopic techniques are well established [3]. Endoscopic techniques have certain limitations: they offer two-dimensional visualization without significant magnification of the surgical field and require one hand to hold the endoscope, leaving the surgeon with only one hand available for manipulation. This restricts the possibility of bimanual manipulation of the soft tissues, which is necessary in this type of surgery. This can also limit the possibility of controlling hemorrhages, which are not uncommon, as the base of the tongue is a highly vascularized region. Microscopic techniques, on the other hand, limit the surgeon to a minor surgical field [4]. With this in mind, robotic surgery was developed to improve performance, reduce morbidity and complications, and assist the surgeon by overcoming the disadvantages of microscopic and endoscopic transoral surgery. Consequently, the transoral robotic surgery system offers significant advantages in terms of visualization and instrumentation [4]. This system provides the surgeon with a high-resolution three-dimensional (3D) view of the surgical field, simulating the perspective of being inside the patient’s mouth. The maneuverable telescope provides a wide field of view, and the wrist instruments significantly reduce tremors. The advanced movement scaling and the wrist’s seven degrees of freedom allow for highly precise bimanual tissue manipulation, enabling surgeons to reach and operate on areas that were previously inaccessible through traditional oral approaches. This enhanced dexterity and control significantly improve the accuracy and effectiveness of intricate surgical procedures. Despite the lack of haptic feedback, many experts claim that the superior three-dimensional (3D) vision compensates for this limitation [5]. Because TORS allows access to previously inaccessible areas, it is becoming increasingly popular for the surgical treatment of base of tongue (BOT) disorders. This less invasive approach reduces the comorbidities associated with surgery and is now being used for benign pathologies. Within the oropharyngeal structure, the base of the tongue is of critical importance to swallowing and breathing functions, making pathologies in this area potentially life-threatening. Tongue base masses (TBMs) can cause acute airway obstruction and significant difficulty in feeding, especially in infants and children. The leading causes of TBMs are usually infectious diseases, congenital lesions, and neoplasms. The most common congenital lesions include lingual thyroglossal duct cysts (LTGDCs), lingual thyroid cysts, vallecular cysts, and hemangiomas [4]. A rare developmental anomaly resulting from abnormal embryogenesis, the ectopic thyroid gland, often manifests at the base of the tongue, with a prevalence of 1:100,000 to 1:300,000 [6]. As for infectious pathologies, the BOT is usually not the first site of infection; the larynx or epiglottis are more frequently involved, especially in cases of severe airway obstruction.

However, recurrent tonsillitis may primarily affect the lingual tonsil and present as acute inflammation of the lymphoid tissue at the base of the tongue. Symptoms may include a foreign body sensation, pharyngodynia, dysphagia, fever, and signs of upper respiratory tract infection. Chronic cases tend to present with non-specific symptoms, often leading to an underestimated diagnosis [7,8,9]. Given the benign nature, any treatment should be well-weighed in terms of benefit and local morbidity. The Da Vinci robotic system is proving to be a promising and valid alternative to conventional approaches for the treatment of a range of pathologies, symptoms, and treatment protocols. This article aims to provide an overview of the treatment of these benign pathologies with this innovative instrument and to explain the decision-making process underlying the selection of specific surgical approaches.

## 2. Materials and Methods

### Search Strategy

A systematic search for articles published in the PubMed and Google Scholar databases between January 2003 and December 2023 was performed using the following combined search query (robot OR sleep OR apnea OR syndrome) AND (robot OR tongue OR base). Our search methodology followed the Preferred Reporting Items for Systematic Reviews and Meta-Analyses (PRISMA) guidelines [10].

This research direction was chosen as there are few publications addressing the treatment of benign pathologies of the oropharynx by transoral robotic surgery (TORS) despite the increasing popularity of this surgical technique. The primary inclusion criteria were the use of robotic surgery and a focus on pathologies affecting a specific part of the oropharynx: the base of the tongue (BOT). Duplicate publications were excluded, and the selected articles were screened using general exclusion criteria: Meta-analyses or reviews, lack of full text, languages other than Italian or English, and articles that addressed related topics but needed to be more specific. Subsequently, all abstracts were reviewed and further screened by one author (E.Z.). Papers related to non-TORS surgery, cancer, obstructive sleep apnea syndrome (OSAS), and removal of foreign bodies from the BOT were excluded from this study. The selection process and PRISMA flowchart are summarized in Figure 1.

## 3. Results

After screening, we selected abstracts for full-text analysis of articles. Three hundred and seventy-three articles were found on Pubmed and Google Scholar. After applying the inclusion and exclusion criteria, only 16 articles were considered relevant and thus included in our work. We reviewed the full texts and compiled key information from each article, including authorship, year of publication, country, type of study, sample size, diagnosis, and main results. The articles are summarized in Table 1.

Due to the limited availability of research on the TORS approach, we found that most articles consist mainly of case series or case reports.

Petruzzi et al. described the TORS approach as a valid alternative to the conventional transoral approach for schwannomas at the base of the tongue. They achieved complete surgical removal while preserving tongue function and minimized postoperative morbidity in their patient [11]. In relation to benign pathology, Montevecchi et al. described a case series in which reduction of the base of the tongue and supraglottoplasty using the TORS approach was an effective option to aid decannulation in patients with severe hypertrophy of the base of the tongue [12].

Most papers analyzed cases of ectopic cysts of the thyroid gland and thyroglossal duct [13,14,15]. A retrospective case series by Johnston et al. presented the TORS approach for cysts of the thyroglossal duct in pediatric patients, a new area of practice that is currently only performed in a limited number of specialized centers. The authors have successfully treated all cases by transcervical en bloc removal of the central hyoid bone together with the tract leading to the lingual lesion. They also argue that the recurrence of thyroglossal duct cysts (TGDC) after a conventional transcervical modified Sistrunk procedure may be due to an unrecognized component at the base of the tongue. Therefore, the authors emphasize the importance of clearly defining the extent of tongue base involvement in both primary lingual lesions near the hyoid bone and recurrent TGDC [16].

Regarding TORS treatment of ectopic thyroid, all authors reviewed agreed that it is a tolerable and intuitive procedure. May et al. were the first to describe TORS, while Curtis et al. emphasized its efficacy in relieving symptoms of sleep apnea associated with a lingual thyroid gland [17,18].
diagnostics-15-00005-t001_Table 1Table 1Selected articles in the review with key characteristics/features.Author and YearCountryType of StudySample SizeRoboticSystem UsedDiagnosisOutcomeVenkatakarthikeyan, [1] 2020IndiaCase Series1Da VinciEpidermoid cyst of the tongue basePediatric robotic surgery is a safe, feasible, and effective technique that can be performed in tertiary care centers by experienced surgeons.Kayhan, [4] 2017TurkeyRetrospective8Da VinciCongenital Tongue Base MassesTORS provides better exposure, visualization, and access than other transoral techniques for children. TORS is a viable, safe, and effective technique for pediatric patients. TORS is superior to traditional open and other transoral procedures when considering the functional and esthetic results achieved in this seriesDi Luca, [7] 2020ItalyRetrospective84Da VinciLingual TonsillitisThe health status of patients after TORS for RLT has improved, the incidence of acute inflammation has decreased significantly, and the intake of medication has been considerably reduced. Excellent results were reported in terms of postoperative quality of life and swallowing function.Kayhan, [19] 2013TurkeyCase Report1Da VinciThyroglossal duct cystBased on this case, TORS was found to be a reliable and effective treatment method for lingual TDC. As a result of its low morbidity and surgical advantages, TORS may represent an alternative method for lingual TDC treatment. Fong, [20] 2018AustraliaCase Report1Da VinciThyroglossal duct cystThe use of transoral robotic surgery provides superior access to the lingual TGDC, enabling complete excision to prevent recurrence. A TORS approach to the tongue base also has favorable post-surgical outcomes, with a rapid patient recovery.Turhan, [21] 2019TurkeyCase Report1Da VinciThyroglossal duct cystSurgical treatment of lingual TDC in an infant patient can be performed successfully with a transoral robotic approach and minimal risk of complication. However, further studies are strongly needed to confirm the safety of robotic surgery in the pediatric population.Johnston, [22] 2021USACase Series2Da VinciThyroglossal duct cystThis is the first published description of a purely transoral approach for the excision of a lingual thyroglossal duct cyst (TGDC) and the central portion of the hyoid bone. This contributes to the limited literature on performing this type of surgery robotically and on the application of transoral robotic surgery (TORS) in pediatric patients. Further studies are needed to confirm its safety and to determine whether the outcomes are comparable to traditional approaches.Petruzzi, [11] 2020ItalyCase report1Da VinciTongue base schwannomaTORS has proven to be a valid alternative to the usual transoral approach in the treatment of tongue-based schwannomas. Complete surgical removal of the tumor was achieved with preservation of tongue function and low postoperative morbidity.Montevecchi, [12] 2017ItalyCase series4Da VinciPatient with tracheostomyTORS appears to be an effective option to support decannulation in patients with severe hypertrophy of the BOT and flaccid epiglottis. Dallan, [13] 2013ItalyCase Report1Da VinciEctopic ThyroidAbsolute tolerability of the procedure and its intuitiveness, in contrast to any external procedure.Filarski, [14] 2021USACase Report1Da VinciEctopic ThyroidThe authors report a rare and unexpected manifestation of an ectopic hemorrhagic lingual thyroid gland triggered by alcohol ingestion. They recommend multidisciplinary coordination to secure the airway, embolize the affected lingual vessels, and resect the lesion using transoral robotic surgery.Kimple, [15] 2012USACase Report1Da VinciThyroglossal duct cystThe first report is a TORS resection of an LTGDC. This rare presentation of a TGDC is well suited to resection using the TORS approach. Johnston, [16] 2023USARetrospective Case Series7Da VinciThyroglossal duct cystTORS in pediatric patients remains a novel domain. Currently, a limited number of specialty centers are gaining experience and attempting to broaden surgical indications.Curtis, [17] 2021UKCase Report1Da VinciEctopic ThyroidAlthough lingual thyroid is a rare cause of sleep apnea, this case demonstrates that transoral robotic surgery is an effective and safe method to achieve curative surgery and life-changing outcomes.May, [18] 2011USACase Report1Da VinciEctopic ThyroidTORS was used to remove an obstructing lingual thyroid gland. The improved visualization and instrument control allow transoral excision of lingual thyroid tissue in cases that previously required a transcervical approach. For this reason, TORS should be included in the treatment armamentarium for the lingual thyroid gland.Montevecchi, [23] 2017ItalyRetrospective10Da VinciLingual TonsillitisResection of the lingual tonsils using the robotic technique appears to be feasible and well-tolerated in patients with lingual tonsillitis. The procedure is safe, easy to learn, and not associated with major complications.

## 4. Discussion

Transoral robotic surgery (TORS) is still a young technique, having been approved for transoral surgery 15 years ago. It is now used worldwide for the treatment of both benign and malignant diseases. The most important applications are still oncological surgery of the oropharynx and the treatment of obstructive sleep apnea syndrome (OSAS). However, given the numerous advantages of this technique, it is being used more and more frequently for various pathologies. In this context, it is important to carefully weigh the risks and benefits before recommending a particular surgical approach to a patient [24].

The aim of this review was to highlight the treatment of benign pathologies of the base of the tongue, with the exception of the treatment of OSAS. Most of the selected articles (13 out of 16) were case reports, indicating that we are still in the early stages of using this approach.

A new finding is that TORS also appears to be a safe procedure in the surgical treatment of children. Kayhan et al. were the first to describe the use of TORS in the pediatric population and reported excellent results with minimal complications [4]. The youngest patient treated was a 2-month-old infant with a lingual thyroglossal duct cyst, and the size of the Da Vinci instruments presented a challenge [19]. The primary goal in this patient was rapid resolution of respiratory distress while minimizing morbidity because in pediatric patients, lingual thyroglossal duct cysts can cause respiratory distress and, if left untreated, can lead to sudden death. Kayhan points out that conventional procedures such as the Sistrunk technique can lead to complications, especially when performed as emergency procedures in pediatric patients, including wound complications, bleeding, infection, and airway edema. Although airway edema is a concern, complete transoral removal or marsupialization of the cyst can effectively reduce morbidity by avoiding dissection of the neck, base of the tongue, hyoid bone, and suprahyoid structures [19].

In benign pathologies of the base of the tongue (BOT), we can distinguish between congenital and acquired conditions. Based on the results of our study, the first group includes congenital BOT masses, which mainly affect the pediatric population, as well as ectopic thyroid glands, dermoid cysts, and thyroglossal cysts. Acquired pathologies include benign tumors, such as the base of tongue schwannomas, and infectious diseases, lingual tonsillitis [23]. In addition to the commonly described benefits, TORS has also been shown to be effective in preventing the recurrence of BOT procedures. It is well known that incomplete surgical resection of certain pathologies, such as thyroglossal duct cysts, carries a high risk of recurrence. Kayhan, Fong, Turhan, et al. reported no cases of recurrence after resection with TORS [19,20,21]. In addition, Johnston et al. described the advantages of the post-hyoid space (PHS) in detail. As it is bounded anteriorly and laterally by the hyoid bone, posteriorly by the thyroid membrane, and superiorly by the vallecula, the PHS could theoretically also provide transoral access to anterior cervical masses [22]. Johnston emphasized that the midline approach through transoral robotic surgery allows the surgeon to safely remove the middle third of the hyoid bone without complications. With this approach, the post-hyoid space provides a corridor for the surgeon that is relatively free of blood vessels and lymphatics, allowing access to the anterior region of the neck. Airway management is an important and frequently discussed topic, especially in BOT surgery and particularly in the context of TORS. As shown in Table 2, 11 out of 16 authors did not perform a tracheostomy in their patients. Even Montevecchi et al. have shown how TORS can prevent decannulation failure by intervening at the base of the tongue [12]. In the study by Kumar et al., tracheostomies were only performed during the first year of using the Da Vinci system. This suggests that the need for a tracheostomy may depend on the extent and duration of the procedure. As surgical skill and experience increased, the duration of the procedure decreased, resulting in less postoperative edema of the floor of the mouth or larynx [25]. This topic remains controversial, with treatment approaches varying widely, particularly for benign head and neck pathologies, with the exception of obstructive sleep apnea syndrome (OSAS). In fact, there are significantly more studies in the literature on tracheostomy as part of the TORS approach in patients with malignant disease or OSAS. In a single-center review, Kumar et al. even classified tracheostomy as a complication of the TORS procedure comparable to bleeding and aspiration pneumonia [25]. In contrast, some authors contextualize tracheostomy as a necessary intervention to support patients during cancer treatment, similar to percutaneous endoscopic gastrostomy (PEG) [26]. This suggests that there is no consensus on this issue but that it is certain that multidisciplinary management is required and must be discussed prior to surgical planning. From our analysis, TORS is a generally safe procedure. Kumar et al. have published a review of complications associated with TORS, covering both benign and malignant pathologies. Their results indicate a decreasing trend in the incidence of complications, suggesting the existence of a learning curve where increasing surgeon experience correlates with fewer complications. In terms of morbidity associated with surgery and hospitalization, the less invasive TORS approach to BOT mitigates the discomfort typically associated with open surgical approaches [25]. Given the benign nature of the lesions, any proposed treatment must be carefully considered in terms of potential functional impairment [27]. This review also outlines the limitations of Transoral Robotic Surgery (TORS) for base of tongue procedures. Specifically, there is a limited amount of research available on the use of TORS for benign pathologies of the base of the tongue. Most of the existing studies are case reports or small case series. Another important consideration is the cost and preparation time, which are significant factors when resources are limited. The expense and time needed for preoperative preparation with TORS may not always result in better outcomes, particularly for benign pathologies. Controlling hemorrhages can be challenging with TORS, as the base of the tongue is a highly vascular region. When dealing with benign lesions, it is essential to provide the appropriate surgical indication. Although benign lesions have become more accessible with TORS, this does not necessarily justify the need for surgery. In some cases, this can result in bleeding that may be life-threatening and could necessitate a tracheostomy, which increases patient morbidity.

## 5. Conclusions

Compared to other transoral techniques, TORS offers better exposure, visualization, and access to the oropharynx, especially the base of the tongue, even in benign pathology. TORS should be considered a viable, safe, and effective technique that is superior to traditional open and other transoral procedures in terms of functional and aesthetic results. In the future, TORS could become the gold standard technique for operations at the base of the tongue. However, a cost–benefit analysis needs to be performed. It is well known that it is a costly procedure that not all healthcare systems can afford. There is a significant difference between the public healthcare systems in Europe and the private healthcare systems in the United States and Asia. Because of this limitation, the considerable potential of this procedure is not fully realized. Prospective, randomized studies with large samples are needed to demonstrate this with sufficient statistical credibility.

## Figures and Tables

**Figure 1 diagnostics-15-00005-f001:**
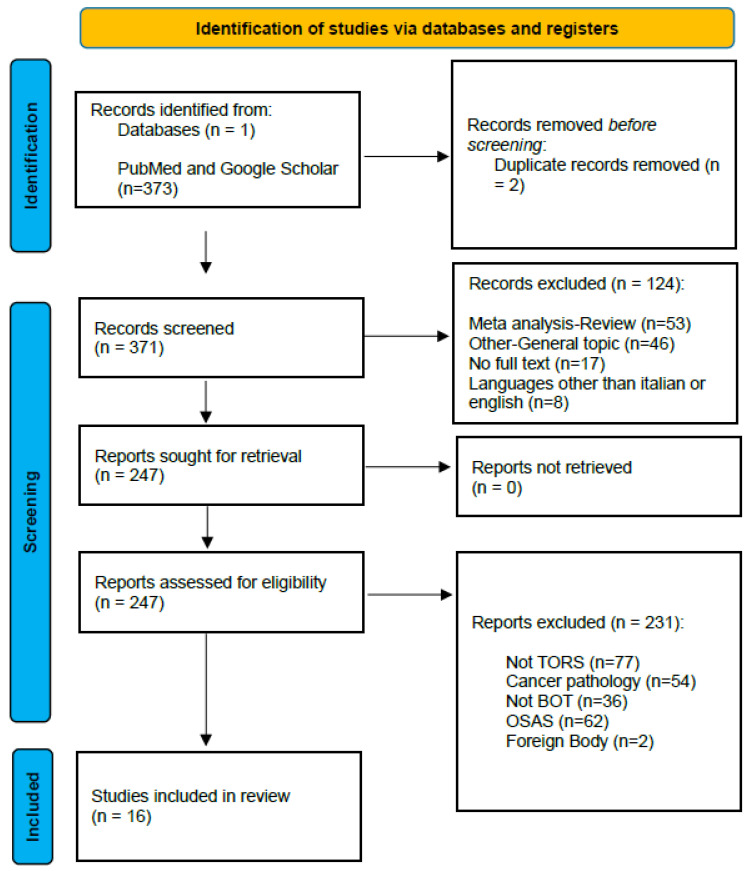
PRISMA.

**Table 2 diagnostics-15-00005-t002:** Need of tracheostomy in the analyzed articles.

Author	Diagnosis	Sample Size	Tracheostomy
Venkatakarthikeyan, [1] 2020	Epidermoid cyst of the tongue base	1	No
Kayhan, [4] 2017	Congenital Tongue Base Masses	8	No
Di Luca, [7] 2020	Lingual Tonsillitis	84	No
Kayhan, [19] 2013	Thyroglossal duct cyst	1	No
Fong, [20] 2018	Thyroglossal duct cyst	1	No
Turhan, [21] 2019	Thyroglossal duct cyst	1	No
Johnston, [22] 2021	Thyroglossal duct cyst	2	1 patient No; 2 patient Yes (for other reasons)
Petruzzi, [11] 2020	Tongue base schwannoma	1	Yes
Montevecchi, [12] 2017	Patient with tracheostomy	4	TORS use for decannulation
Dallan, [13] 2013	Ectopic Thyroid	1	Yes
Filarski, [14] 2021	Ectopic Thyroid	1	Yes
Kimple, [15] 2012	Thyroglossal duct cyst	1	No
Johnston, [16] 2023	Thyroglossal duct cyst	7	No
Curtis, [17] 2021	Ectopic Thyroid	1	No
May, [18] 2011	Ectopic Thyroid	1	No
Montevecchi, [23] 2017	Lingual Tonsillitis	10	Yes

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
