# Peer review of "The Role of TORS in the Management of Benign Pathology of the Base of Tongue: A Systematic Review"

_diagnostics, 2024, doi:10.3390/diagnostics15010005_

Round 1
Reviewer 1 Report
Comments and Suggestions for Authors
Review on a niche topic listing case reports/series on the subject, without delving into the characteristics that make the Tors approach peculiar on the base of tongue. It would be useful to better specify for the reader which features of the BOT make a transoral approach so difficult and why the robot overcomes the difficulties of working a deep and curved structure that is difficult to expose properly and resect.
I would dwell more on complications, in particular the risk of postoperative haemorrhage even days after surgery (>10 days) that can put the patient's life at risk (for a benign pathology in this case) and the protective role of tracheotomy in this regard.
I don't think that having an assistant holding an endoscope is a sufficient reason to declare TORS superior to endoscopic surgery (also considering the costs of TORS and the limitations related to exponibility with robotic arms). It is useful to elaborate...
Author Response
Dear Reviewer, thank you for your feedback. We appreciate your suggestions as they help to enhance the quality of our work. The changes you requested have been made.
Review on a niche topic listing case reports/series on the subject, without delving into the characteristics that make the Tors approach peculiar on the base of tongue. It would be useful to better specify for the reader which features of the BOT make a transoral approach so difficult and why the robot overcomes the difficulties of working a deep and curved structure that is difficult to expose properly and resect.
R: Thank you, we included phrases that explain the challenges of accessing this region.
I would dwell more on complications, in particular the risk of postoperative haemorrhage even days after surgery (>10 days) that can put the patient's life at risk (for a benign pathology in this case) and the protective role of tracheotomy in this regard.
R: In terms of benign pathology, tracheostomy may cause more harm than benefit. Table 2 reflects these considerations in the analyzed articles.
I don't think that having an assistant holding an endoscope is a sufficient reason to declare TORS superior to endoscopic surgery (also considering the costs of TORS and the limitations related to exponibility with robotic arms). It is useful to elaborate...
We have elucidated the reasons why TORS is considerably superior to the endoscopic technique. The primary rationale lies in the enhanced mobility and range of movement provided by the two robotic arms.
Thank you for your feedback.
Sincerely,
The authors
Reviewer 2 Report
Comments and Suggestions for Authors
The manuscript is quite well written and is very relevant in the field of Oral & maxillofacial surgery.However,the authors need to address a few points as follows:
1.There are a few grammatical errors strewn across the manuscript and a thorough written english language correction needs to be done.
2.The authors need to elaborate on the limitations of TORS
3.It would be desirable if the authors had compared TORS to other modes of surgery in treatment of pathologies of BOT in the discussion section
Comments on the Quality of English Language
The manuscript is quite well written and is very relevant in the field of Oral & maxillofacial surgery.However,the authors need to address a few points as follows:
1.There are a few grammatical errors strewn across the manuscript and a thorough written english language correction needs to be done.
2.The authors need to elaborate on the limitations of TORS
3.It would be desirable if the authors had compared TORS to other modes of surgery in treatment of pathologies of BOT in the discussion section
Author Response
Dear Reviewer,
We sincerely thank you for your feedback. We have followed your suggestions to improve the quality of our paper.
1.There are a few grammatical errors strewn across the manuscript and a thorough written english language correction needs to be done.
R: We have accordingly reviewed the manuscript and corrected the grammatical errors
2.The authors need to elaborate on the limitations of TORS
R: Added as requested
3.It would be desirable if the authors had compared TORS to other modes of surgery in treatment of pathologies of BOT in the discussion section
References to endoscopic and open approaches have been added. However, we do not consider these procedures to be effective treatments for benign conditions of the base of the tongue. The endoscopic approach provides limited visualization of the area and is not considered safe. On the other hand, open approaches in benign pathology can be overly invasive. Conducting a study in this manner could be relevant in oncology; however, a multicentric study is required to obtain a sufficient number of cases for the findings to be significant.
We sincerely appreciate your valuable suggestions.
Yours faithfully,
The Authors.